# Chitosan-Based Structural Color Films for Humidity Sensing with Antiviral Effect

**DOI:** 10.3390/nano14040351

**Published:** 2024-02-13

**Authors:** Darya Burak, Dong-Chan Seo, Hong-Eun An, Sohee Jeong, Seung Eun Lee, So-Hye Cho

**Affiliations:** 1Materials Architecturing Research Center, Korea Institute of Science & Technology, 5 Hwarang-ro 14-gil, Seongbuk-gu, Seoul 02792, Republic of Korea; burak_dasha@kist.re.kr (D.B.); ahe9930@kist.re.kr (H.-E.A.); soheejeong@kist.re.kr (S.J.); 2Department of Nanomaterial Science and Engineering, Korea University of Science and Technology, 217 Gajeong-ro, Yuseong-gu, Daejeon 34113, Republic of Korea; 3Research Animal Resources Center, Korea Institute of Science & Technology, 5 Hwarang-ro 14-gil, Seongbuk-gu, Seoul 02792, Republic of Korea; cucool@kist.re.kr; 4School of Biosystem and Biomedical Science, College of Health Science, Korea University, 145 Anam-ro, Seongbuk-gu, Seoul 02841, Republic of Korea; 5Department of Materials Science and Engineering, College of Engineering, Korea University, 145 Anam-ro, Seongbuk-gu, Seoul 02841, Republic of Korea

**Keywords:** chitosan, thin film, structural color, surface decoration, antiviral and antibacterial film, humidity sensing

## Abstract

This scientific investigation emphasizes the essential integration of nature’s influence in crafting multifunctional surfaces with bio-inspired designs for enhanced functionality and environmental advantages. The study introduces an innovative approach, merging color decoration, humidity sensing, and antiviral properties into a unified surface using chitosan, an organo-biological polymer, to create cost-effective multilayered films through sol-gel deposition and UV photoinduced deposition of metal nanoparticles. The resulting chitosan films showcase diverse structural colors and demonstrate significant antiviral efficiency, with a 50% and 85% virus inhibition rate within a rapid 20 min reaction, validated through fluorescence cell expression and real-time qPCR (polymerase chain reaction) assays. Silver-deposited chitosan films further enhance antiviral activity, achieving remarkable 91% and 95% inhibition in independent assays. These films exhibit humidity-responsive color modifications across a 25–90% relative humidity range, enabling real-time monitoring validated through simulation studies. The proposed three-in-one functional surface can have versatile applications in surface decoration, medicine, air conditioning, and the food industry. It can serve as a real-time humidity sensor for indoor and outdoor surfaces, find use in biomedical devices for continuous humidity monitoring, and offer antiviral protection for frequently handled devices and tools. The customizable colors enhance visual appeal, making it a comprehensive solution for diverse applications.

## 1. Introduction

Throughout history, nature has consistently provided lessons on sustaining and advancing the human-built environment by revealing its fundamental designs and mechanisms. Biomimicry, drawing inspiration from nature’s bio-inspired forms, shapes, and processes, holds undeniable advantages over artificially designed materials of purely chemical origin [1,2,3,4,5]. Bio-inspired micropatterns, for instance, require fewer chemical substances, such as the opal colors resulting from light interacting with 3D microstructures made of air voids [6]. Additionally, materials in nature-inspired structures are generally less harmful and polluting, with most bio-polymers being ecologically friendly and non-toxic [7,8].

These compelling advantages fueled a growing interest in bio-inspired functional surfaces at both scientific and industrial levels. Functional surfaces, possessing properties like adhesion [9], self-cleaning [10], superhydrophobicity [11], anti-biofouling [12], wear-resistance [13], or energy harvesting [14], were extensively studied, showing promising real-life applications [15,16,17]. For instance, research into Van der Waals forces inspired the design of a microscale fibrillar adhesive with gecko-like strength [18], and structural color tuning [19,20,21] led to innovations like a soft robot capable of color camouflage [22].

Despite extensive research, most studies focus on a single function of a fabricated surface, with relatively few exploring multifunctionality [23,24,25,26,27,28]. This study aims to broaden the scope by combining three diverse functions into a bio-inspired surface. We propose a multifunctional surface with tunable color decoration properties based on structural colors whilst enabling in situ color change for humidity sensing and efficient virus and bacteria elimination via intrinsic chitosan antimicrobial properties. This three-in-one functional surface can find applications as a film in interior decoration [29,30], biomedicine [31], food processing [32,33], and air conditioning [34], addressing the need for humidity monitoring and active protection against pathogens [35,36,37,38], while enhancing the visual appeal of the functional surfaces. For example, our proposed functional surface can be utilized in biomedical devices, providing real-time humidity monitoring through vivid color change and antimicrobial protection [39] on the external surface of frequently handled devices and tools. In air conditioning applications, it can aid in monitoring ambient humidity levels, removing indoor airborne toxins such as formaldehyde [40], and adding an aesthetic appeal to interiors.

The functional surface features an organo-biological polymer—chitosan—as a functional layer for multilayered films. In the realm of their applications, including food packaging, the functional films developed in this study are classified as intelligent films [41]. Unlike active packaging, these films do not have direct contact with the food; instead, they serve to monitor and sense the conditions of the packaged food and its environment. Compliant with the European Union’s regulatory framework for materials in contact with food, particularly EC 450/2009 and Regulation 1935/2004 (pp. 4–17), our chitosan-based intelligent films, possessing biodegradability and antimicrobial properties, effectively meet both regulatory and industry requirements [42,43]. Chitosan is a biopolymer produced from chitin, an abundant natural polymer present in the exoskeleton of shrimp, crabs, and other crustaceans [15]. Chitosan is a remarkable polymer with an impressive scope of versatile and diverse applications: as of 2022, chitosan covers up to 1 million citations on Google Scholar, with its biodegradability, biological compatibility, antimicrobial and antioxidant, and non-toxic nature paving its numerous applications [44,45,46].

In our study, the following three primary properties of chitosan were utilized. Firstly, chitosan served as an insulating material in the design of multilayered films [47], where the resulting structural coloration arises from interference effects within the constituent layer [48], contributing to the esthetic appeal and color diversity of the surface. Secondly, the chitosan films were doped with metallic nanoparticles [49] through a photodeposition process. Metallic nanoparticles, such as silver, copper, and zinc, possess antibacterial and antiviral properties [50,51], which would enhance the chitosan’s intrinsic capability to combat pathogens while saturating the original chitosan film color via enhanced interference [52,53]. Furthermore, silver was intentionally chosen among metallic nanoparticles due to its lowest reduction potential (0.8 eV) [54], facilitating efficient photoreduction in the fabrication process for silver nanoparticles in this study. Lastly, a unique property of chitosan—its tendency to swell by absorbing moisture from the air, attributed to its hydrophilic nature owing to amine (NH_2_) and hydroxyl (OH^−^) groups in its structure [55]—in response to changes in relative humidity was complementarily utilized. This inherent feature allowed for the design of a humidity sensor, wherein chitosan swelling induces variation in film thickness, subsequently leading to a discernible color change [46], thus allowing the monitoring of humidity changes in real-time.

In essence, the multifunctional properties of the three-in-one functional surface stem from the synergistic contributions of both chitosan and silver nanoparticles. Chitosan serves as the foundational material, providing structural coloration, antiviral properties, and humidity responsiveness. The introduction of Ag nanoparticles complements these functionalities by enhancing antiviral properties and saturating coloration through intensified interference effects. The interplay between these two factors results in the comprehensive performance of the multifunctional surface.

## 2. Materials and Methods

### 2.1. Materials

Stainless steel (SUS) substrates (type 304, 2 cm × 2 cm, thickness 0.1 cm) were customized and supplied by Daihan Scientific Co., Ltd. (Seoul, Republic of Korea). Chitosan (from shrimp shells, ≥75% (deacetylated)) was purchased from Sigma Aldrich (Darmstadt, Germany). Acetic acid (glacial, ≥99.7%), ethanol (EtOH), isopropanol (IPA), and silver nitrate (AgNO_3_) were obtained from Daejung Chemicals & Metals Co., Ltd. (Daejung, Republic of Korea) and Alfa Aesar (Ward Hill, MA, USA), respectively. These compounds were used without additional purification. Deionized water (DI) (4.6 MΩ·cm) was prepared using the Millipore Milli-Q laboratory water system. The commercial Ag film employed in this study was manufactured domestically and consisted of Ag particles embedded in polyethylene (PE), hereafter referred to as antibacterial PE/Ag.

### 2.2. Preparation of Chitosan Precursor Solution

The preparation method of the chitosan solution was adopted from [46] with some modifications. Briefly, a 1 wt.% chitosan solution was prepared by dissolving 0.41 g of chitosan powder in 40 mL DI water containing 1.5 wt.% acetic acid. The mixture was stirred and heated overnight at 80 °C with a stirring speed of 1000 rpm. Subsequently, it was centrifuged at 8000 rpm and 25 °C for 30 min. After centrifugation, the solution was consistently heated at 80 °C on a hot plate to maintain a moderate viscosity suitable for spin-coating.

### 2.3. Fabrication of Chitosan and Ag/Chitosan Films

Before film deposition, SUS and silicon substrates were first ultrasonically cleaned in EtOH, IPA, and DI in sequence and then dried under a stream of argon. Further removal of surface contaminants was achieved by the UV/ozone cleaning procedure in a UV Ozone Cleaner UVC-30S (Jaesung Engineering Co. (Anyang-city, Republic of Korea)).

Chitosan films were fabricated by spin-coating the chitosan precursor solution onto the pretreated substrates at speeds from 1000 to 5000 rpm. The number of spin-coating cycles, ranging from one to three, was determined based on the desired color for each sample. Following spin-coating, the polymer films were dried at 80 °C for 1 h.

Silver deposition onto the chitosan surface was achieved by a facile one-pot method. A total of 10 mM AgNO_3_ solution in DI/ethanol (1:1) was prepared and maintained at a constant concentration throughout this study. The solution was then applied drop-wise onto the chitosan film surface and exposed to UV-C irradiation (Sankyo Denki (Kanagawa, Japan), 253.7 nm, 15 W) for approximately 7 min. Chitosan is a relatively robust polymer, which can sustain temperatures up to 200 °C [56] and UV exposure for a short period of time [57]. Herein, the deposition of silver nanoparticles was facilitated by the in situ reduction of Ag^+^ from AgNO_3_ under UV-C irradiation, and the deposition of silver was evidenced by an increased color saturation of the chitosan film.

### 2.4. Antiviral Tests

The details of the antiviral assays can be found in the Method section of the Appendix A. The procedures were adapted from our previous study [52] with slight modifications.

### 2.5. Humidity-Induced Color Change Experiments

Initially, a chitosan/Ag film was fabricated under ambient conditions, and the relative humidity (RH) was measured at 45% using a humidity sensor (Morning Glory (Seoul, Republic of Korea)). The color and corresponding reflectance spectrum were recorded at this stage. Subsequently, the film was transferred to a chamber equipped with a transparent window within a temperature and humidity control chamber (S-TH31, SERIMA).

In this controlled environment, various RH levels (25, 65, 75, and 90%) were introduced to the chamber, with RH being adjusted by regulating the supply of water vapor. Appendix A displays a schematic diagram of the humidity-generating system. Color changes were observed within the chamber to accurately capture RH values corresponding to specific colors. The film was placed inside the chamber once a specific set RH value was stabilized. The color change was then monitored every 1 min for 10 min. The film was then taken out from the chamber, and the reflectance spectra of the colors were measured. To minimize measurement errors and maintain consistent colors, the humidity chamber was positioned adjacent to the spectrophotometer during reflectance spectra measurements. This approach ensured precision in recording color changes under controlled humidity conditions.

### 2.6. Characterization

The elemental chemical states in the fabricated films were determined using grazing incidence X-ray diffraction (D8 Advanced diffractometer (Bruker Corporation (Billerica, MA, USA)). The plain view of both the chitosan and Ag/chitosan films, as well as cross-sections of the chitosan films with varying thicknesses, were examined using field-emission scanning electron microscopy (FESEM) (FEI Inspect F50 (Oregon, OR, USA)). The SEM samples were imaged at 10 and 15 kV for the chitosan and Ag/chitosan films, respectively. The voltage was reduced for the chitosan film due to its rapid damage and degradation under beam exposure. The TEM samples were prepared via focused-ion-beam (FIB) and analyzed via transmission electron microscopy (Cs-TEM) (Neo ARM/Jeol (Tokyo, Japan)). Depth profiles of the elements in the films were determined using time-of-flight secondary ion mass spectroscopy (ToF-SIMS 5, ION-TOF(Münster, Germany)).

Reflectance spectra were measured using a spectrophotometer (Konica Minolta CM 3600A (Tokyo, Japan)) equipped with a white xenon light source with a 4 nm diameter beam. The detector and incident light beam were positioned at 8° from the surface normal. Simulated reflectance spectra were generated for comparison with the measured data using Bruggeman’s EMA (effective medium approximation), as previously described in [58]. In the analysis of the Ag/chitosan/SUS film described in the Results and Discussion section, the Ag layer in this study was regarded as a combination of Ag/air and Ag/chitosan composites. The complex refractive indices of the Ag/air and Ag/chitosan composites, as well as the subsequent reflectance spectra and color simulations, were derived following a detailed process described in our previous study [52]. The weight fractions (*f*) of silver in the Ag/air and Ag/chitosan layers, which were used for the simulation, were 0.21 and 0.19, according to SEM plain-view and TEM EDS cross-section measurements [52].

## 3. Results and Discussion

### 3.1. Characterization of Chitosan- and Ag/Chitosan-Derived Films

To gain insights into the structural and antimicrobial properties of the fabricated chitosan and silver-deposited chitosan films, a thorough characterization was conducted. The GIXRD analysis revealed multiple peaks between 27° and 54°, indicating the amorphous nature of chitosan (Figure 1a). A noticeable hump at 10° was attributed to air-scattering during GIXRD measurements. Notably, distinct diffraction lines at 39.0°, 44.8°, and 64.9° were identified in the GIXRD pattern, signifying the presence of face-centered cubic (fcc) silver (111, 200, and 220) (JCPDS file no. 04-0783) in the fabricated films.

The surface morphology observed through SEM illustrated a uniform and homogeneous chitosan film surface (Figure 1b). The Ag-deposited chitosan film, as depicted in Figure 1c, exhibited a dense distribution of silver nanoparticles, confirming successful silver deposition through UV light-assisted photodeposition. Importantly, the silver nanoparticles were evenly dispersed without noticeable agglomeration on the chitosan surface. EDS and plain-view mapping analyses (Appendix A) corroborated the homogeneous distribution of all elements in the films. The surface silver weight fraction of 0.21, utilized for the color simulation, was obtained from the SEM EDS data presented in Appendix A.

TEM images revealed a multilayer structure in the Ag/chitosan film, with silver nanoparticles present both on the chitosan film surface and embedded within the chitosan layer (Figure 2 and Appendix A). The weight silver fraction within the chitosan layer (0.19) was obtained from the TEM EDS data presented in Appendix A. Since the silver deposition process involved immersing the dried chitosan film into an aqueous solution of AgNO_3_, the chitosan film, being hydrophilic (Appendix A), absorbed the silver nitrate solution, which subsequently caused the film to swell and lead to a uniform distribution of the silver precursor within the chitosan film. This absorption phenomenon facilitated the penetration of AgNO_3_ into the chitosan film, resulting in the formation of embedded silver nanoparticles. This hypothesis was validated by SIMS analysis, showing a prominent silver peak up to ~6 nm from the surface, corresponding to surface silver nanoparticles. Beyond 6 nm, the silver peak intensity decreased yet maintained at a relatively steady level, accompanied by an emergence of the steady plateaued carbon, oxygen, and hydrogen peaks, indicative of chitosan. Consequently, the Ag layer was identified as consisting of two composites: Ag/air, contributing to antiviral and interference coloration effects, and Ag/chitosan, responsible for plasmonic coloration effects. Interestingly, there was an observed size difference between silver nanoparticles in the Ag/air surface and those embedded in the Ag/chitosan film. In the Ag/chitosan layer, the penetration of silver nitrate into the chitosan medium leads to well-distributed silver nuclei and a relatively homogeneous size distribution, preventing processes like Ostwald ripening; nanoparticles, once formed, are effectively trapped within the chitosan matrix, preventing interactions that could lead to further growth. Consequently, the average nanoparticle size remained small, approximately 3–4 nm (as depicted in high-resolution images of Figure 2a). Conversely, in the Ag/air surface composite, small particle redissolution and Ostwald ripening [59] contribute to agglomerated and clustered nanoparticles, explaining their larger size. On the chitosan surface, the average size of silver nanoparticles was measured to be 20–25 nm (Appendix A). These distinct mechanisms in Ag/chitosan and Ag/air provide insight into the underlying processes governing silver nanoparticle formation in each case. Conversely, one can expect the silver nanoparticle size in individual layers will remain uniform (Appendix A, data presented for the Ag/air composite with an average 20–25 nm silver nanoparticle size for all chitosan thicknesses), regardless of variations in the silver/chitosan film thickness, as the silver deposition parameters, specifically the concentration of the silver precursor, were maintained at a constant to ensure the consistent distribution and penetration pattern during the silver deposition process.

### 3.2. Chitosan- and Ag/Chitosan-Derived Structural Color Films

Four representative chitosan films of varying thicknesses—45 nm, 180 nm, 300 nm, and 520 nm—were fabricated in this study by varying rotational speed (from 1000 to 5000 rpm) during spin coating, followed by their measured reflectance spectra comparison with simulated counterparts. The experimental results, as depicted in Figure 3a, indicated a high degree of agreement between the measured spectra and colors and those predicted by simulation. This alignment was further supported by cross-section SEM images, confirming that the fabricated films closely matched the simulated thickness values, albeit slightly thinner.

When silver nanoparticles were introduced into the chitosan films, similarly, the measured reflectance spectra and colors of the Ag/chitosan/SUS films were compared with simulated spectra (Figure 3b). The comparison revealed a satisfactory agreement, although slight discrepancies were noted, especially with the thickest (520 nm) film, primarily attributed to the limitations of the simulation process. The simulation employed Bruggeman’s EMA [58], which relies on bulk dielectric constants and volume fractions, neglecting the effect of silver nanoparticle size, which can potentially contribute to plasmonic coloration effects.

Figure 4 illustrates the successful deposition of silver on the chitosan film, confirmed by studying the structural colors arising from interference effects and presents measured reflectance spectra and corresponding colors for both chitosan/SUS and Ag/chitosan/SUS films across different chitosan layer thicknesses. Chitosan/SUS films exhibited low reflectance dips, resulting in pale and light colors. Conversely, Ag/chitosan/SUS films, with higher absorption due to silver, which induced interference effects [48,52,53] within the multilayered thin film structure, displayed brighter and more vivid colors. Despite a modest red shift in the spectra, indicative of a minor plasmonic effect of silver nanoparticles [60], the overall agreement between measured and simulated results was notable.

The resulting color palette, demonstrated in Appendix A, exhibited moderate saturation across a thickness range of 30 nm to 520 nm. Notably, thickness values of 220 nm, 290 nm, 390 nm, 420 nm, and 520 nm yielded colors with the highest saturation, namely, warm yellow, dark blue, pink magenta, and green, as depicted in the CIELAB (International Commission of Illumination, by which colors are expressed as three values: L* for lightness and a* and b* for four colors of human vision: red, green, blue, and yellow) a*b* coordinate diagram (Appendix A).

### 3.3. Antiviral and Antibacterial Activity of Chitosan and Ag/Chitosan Films

Figure 5a,b demonstrate the results of independent virus infectivity assays, evaluating the fluorescence expression of virus-infected cells on various surfaces: silicon wafer (control), silicon wafer/chitosan, silicon wafer/chitosan/silver films, and a commercial PE/AG antibacterial film for reference (an antiviral assay adopted from our previous study [39]). The fluorescence images (Figure 5a) and the counts of infected cells (Figure 5b, left) show that the Ag-embedded polyethylene film (commercial product) exhibited antiviral activity comparable to the silicon wafer control sample, signifying its limited antiviral performance. In contrast, chitosan and chitosan/Ag films demonstrated promising antiviral efficacy, with rapid inhibition rates of 50% for chitosan and 91% for chitosan/Ag films within the initial 20 min of virus exposure (Figure 5a,b).

Chitosan has long been known for its intrinsic antiviral properties [61,62,63], attributed to its amino (NH_2_) and hydroxyl (OH) functional groups [64,65]. These groups are believed to mediate antiviral activity by positively interacting (amino group) with the negatively charged virus cell membranes, preventing viral entry into host cells and eventually causing disruption of the virus cell, thus impeding replication. Although pure chitosan films showed a notable antiviral effect in this study (50% inhibition), there was room for improvement. This enhancement was achieved by incorporating silver nanoparticles, which modified and amplified the intrinsic antiviral properties of chitosan.

The mechanism of the antiviral effect of silver is still debated; however, it is generally accepted that silver binds to glycoproteins on virus surfaces, leading to the disruption of the virus envelope [66]. In the case of the PE/Ag film, the incorporation of Ag nanoparticles in the medium (PE) hindered consistent direct contact with the virus, reducing its effectiveness. However, in the chitosan/Ag film, the top Ag layer on the chitosan surface facilitated direct contact with the virus, resulting in rapid and efficient binding and, consequently, an improved inhibition rate.

To further investigate the antiviral activity of the films, independent real-time qPCR (quantitative polymerase chain reaction) assays were conducted. Figure 5c illustrates a significant 85% and 95% reduction in lentivirus titer between the control sample and the chitosan and chitosan/Ag films, respectively. Overall, the antiviral performance of chitosan and chitosan/Ag films surpassed that of the PE/Ag films, with the latter showing a significantly lower inhibition rate in comparison.

### 3.4. Humidity-Responsive Color Change in Chitosan Films

Unlike semiconductor-based sensors that rely on electrochemical signals for humidity sensing [67,68], polymeric materials achieve humidity sensing through structural conformation changes triggered by interactions with water molecules, leading to observable color changes, detectable through color spectrum measurements as well as visual observation. For instance, Georgaki et al. introduced a photonic crystal-based humidity sensor with alternating hydrophobic and hydrophilic layers [69], Mergu et al. utilized conjugated polyacetylene for rapid colorimetric response [70], and Kim et al. developed block copolymers with tailored hydroscopic properties covering visible regions in response to humidity variations [71].

Given that the chitosan films generated structural colors based on the thin film interference of a multilayered structure, as discussed in Section 3.1, the fabricated structural colors were further investigated to evaluate their response through color change to relative humidity (RH) variations. Chitosan is well-known for its ability to swell in response to changes in RH by absorbing water [72,73]. Chitosan’s humidity-induced swelling is driven by the formation of hydrogen bonds with its hydrophilic functional groups (amine NH_2_ and hydroxyl OH^−^) [74,75], leading to reversible adsorption and desorption of water molecules within the film. As humidity levels rise, the film undergoes the physisorption of multiple layers of water molecules through weak hydrogen bonding, resulting in a substantial water layer and an increase in film thickness. Higher relative humidity (RH) values (>50%) lead to enhanced moisture absorption, causing significant swelling and the expansion of the chitosan film. Conversely, at lower RH levels (<50%), fewer water molecules are present, resulting in a thinner film (Figure 6a). This leads to changes in the thickness and refractive index of the chitosan layer, causing a shift in the reflectance spectrum [46]. The resulting shift in reflectance peak and corresponding observable color change can serve as a remotely readable estimate (indicator) of humidity level variations, presenting potential applications in sensing mechanisms. To demonstrate the correlation between color and humidity, the chitosan/Ag film of 255 nm at ambient conditions (RH = 45%) was exposed to four different RH levels (25, 65, 75, 90%), with the corresponding observed colors measured via a spectrophotometer. Figure 6a illustrates that as RH increases, the chitosan layer swells, causing a red shift in the spectrum and generating distinctive variations in structural colors (Figure 6b and Appendix A).

The thickness of the chitosan layer at each RH level was estimated by comparing the measured reflectance spectra with simulation data (Figure 6c). The measured and simulated reflectance spectra and corresponding colors were in good agreement. It is worth noting that the simulation process did not take into account the reflectance index change of the chitosan film after absorbing moisture. This omission was justified by the insignificant difference in reflectance indices between chitosan (1.52 [29]) and water (1.33).

A substantial variation in thickness from 230 to 320 nm was observed across the RH range of 25 to 90%. The CIELAB (L*a*b*) color characterization presented in Appendix A indicates a broad color spectrum achievable with the chitosan/Ag film. Consequently, the rapid color change in response to humidity variations allows for a rough estimation of RH, making these films effective instant RH change indicators. Furthermore, the film fabricated in this study exhibited a fast response to color changes within 1 min attributed to the unique geometry of the chitosan/Ag structure. Unlike typical sandwiched multilayered structures [46], the metal nanoparticle-insulator–metal structure in this study, with silver nanoparticles forming a dense yet discontinuous film (vacant air voids), facilitated the penetration of water vapor through the top layer, quickening the color change process. Furthermore, the humidity-exposed films exhibited recovery time within 1 min for RH below 50%, whereas a slightly slower release of absorbed moisture (approximately 2 min) was observed for RH above 50% owing to a larger volume of moisture to be released from within the chitosan film. Additionally, in Appendix A, two chitosan films initially exposed to 90% humidity and then to 25% demonstrate reversible color changes, confirming the films’ ability to return (recover) to their original colors during transitions between high- and low-humidity environments.

## 4. Conclusions

This study introduced a multifunctional bio-inspired surface, merging structural color fabrication, humidity sensing, and antiviral properties. The three primary attributes of chitosan—its role as an insulator, intrinsic antiviral activity, and humidity responsiveness—were strategically harnessed. Chitosan-based structural colors showcased a diverse color palette within the CIELAB color space, demonstrating high agreement between measured and simulated colors. Furthermore, antiviral assays revealed substantial efficiency, with chitosan and chitosan/silver films exhibiting the highest inhibition rates of 85% and 95%, respectively, within the first 20 min of virus exposure according to the real-time qPCR assay. Lastly, the humidity-responsive color change in the chitosan films was validated across a broad relative humidity range of 25% to 95%. Simulation studies revealed a linear increase in the chitosan film thickness with RH, allowing us to estimate the change in RH with varying colors. In conclusion, this bio-inspired surface, utilizing chitosan, can provide a practical solution for a range of applications, from surface decoration to the utilization of antiviral and humidity-sensing films in everyday settings, such as functional surfaces for plastic/metallic containers for food preservation or real-time humidity sensing, pathogen-minimizing films for biomedical devices and tools.

## Figures and Tables

**Figure 1 nanomaterials-14-00351-f001:**
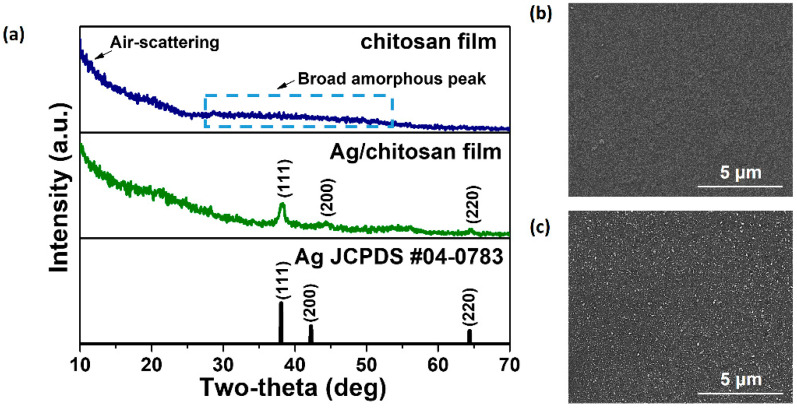
(**a**) GIXRD pattern of chitosan and Ag/chitosan films. SEM plain-view images of (**b**) chitosan and (**c**) Ag/chitosan films.

**Figure 2 nanomaterials-14-00351-f002:**
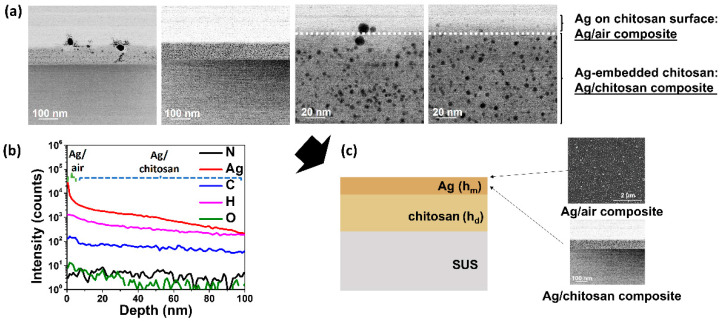
(**a**) TEM images of a cross-section of the Ag/chitosan film. (**b**) SIMS depth profile of a cross-section of the Ag/chitosan film. (**c**) Structure of Ag/chitosan film as per TEM and SIMS analyses.

**Figure 3 nanomaterials-14-00351-f003:**
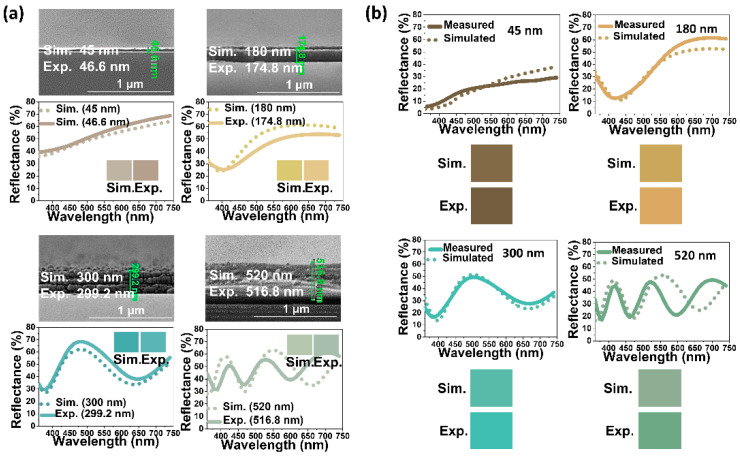
Comparison of measured (experimental) and simulated reflectance spectra and corresponding colors of (**a**) chitosan/SUS and (**b**) Ag/chitosan/SUS films for four representative thickness values (45, 180, 300, and 520 nm) of the chitosan film.

**Figure 4 nanomaterials-14-00351-f004:**
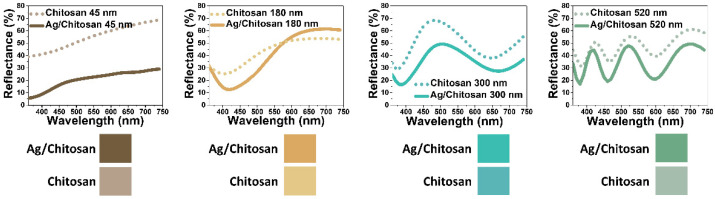
Measured (experimental) and simulated reflectance spectra and corresponding colors of the chitosan/SUS and Ag/chitosan/SUS films for 45, 180, 300, and 520 nm thickness values of the chitosan film.

**Figure 5 nanomaterials-14-00351-f005:**
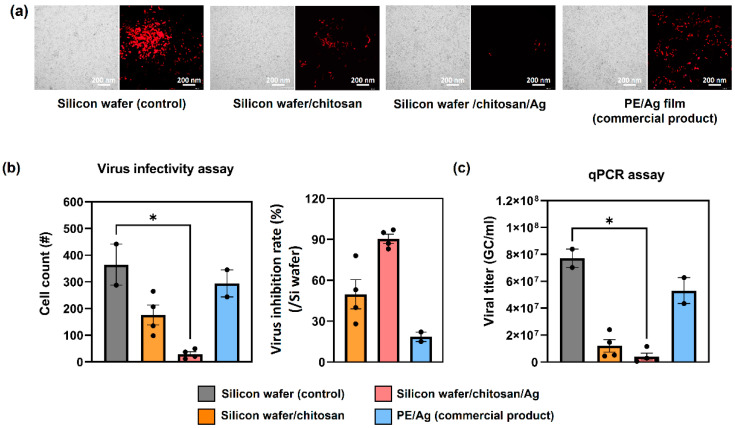
(**a**) Results of fluorescence expression of virus-infected cells assay. Four samples: silicon wafer (control), silicon wafer/chitosan, silicon wafer/chitosan/Ag, and commercial PE/Ag film were exposed to the lentivirus for 20 min. Images on the left represent bright field images of HeLa cells, while images on the right are fluorescence microscopy images of survived lentivirus-infected cells. (**b**) The results of virus infectivity assay. Histogram on the left shows calculated average number of survived infected cells, while histogram on the right shows relative virus inhibition rate compared to control sample (silicon wafer). (**c**) Results of a real-time qPCR assay. The histogram shows the average titration. Both assays were repeated four times (n = 4; black dots on the histograms indicate the individual tests). The significance level is represented as asterisks (* *p* < 0.05).

**Figure 6 nanomaterials-14-00351-f006:**
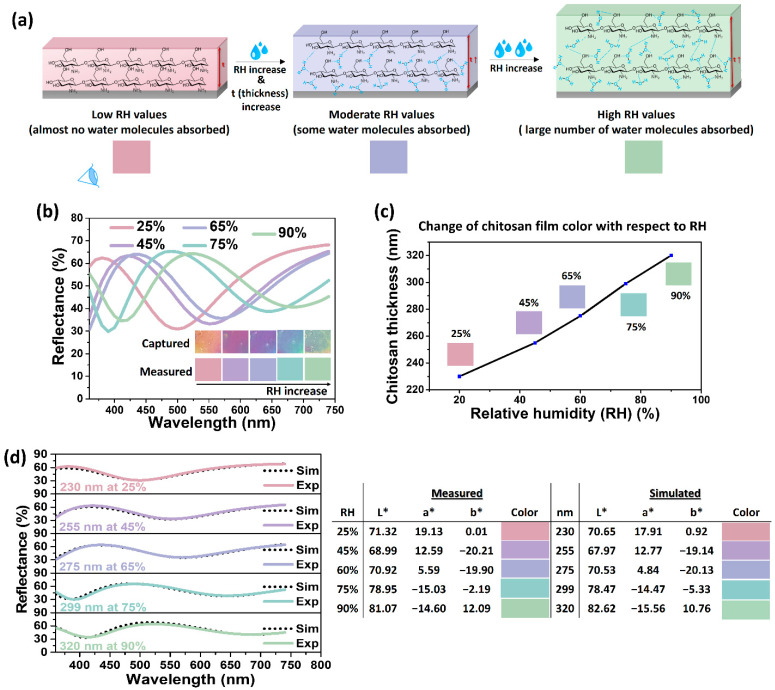
(**a**) Humidity-sensing mechanism based on chitosan swelling (water absorption/desorption) properties. (**b**) Measured reflectance spectra of a chitosan/Ag film under different RH levels (insert demonstrates captured (camera) and measured (spectrophotometer) colors under different RH levels). (**c**) Change of chitosan film thickness with respect to RH estimated from simulation studies. (**d**) Comparison of measured and estimated reflectance spectra under different RH levels.

## Data Availability

Research data are available to interested researchers upon request.

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
