# Peer review of "Chitosan-Based Structural Color Films for Humidity Sensing with Antiviral Effect"

_nanomaterials, 2024, doi:10.3390/nano14040351_

Round 1
Reviewer 1 Report
Comments and Suggestions for Authors
This research introduced a versatile bio-inspired surface that integrates structural color production, humidity sensing, and antiviral properties. In my opinion, lots of doubts should be clarified. The following concerns need to be addressed:
1. From TEM images, the size of Ag nanoparticle at the Ag/air surface appears to be significantly larger than that of the embedded Ag/chitosan. Why? What is the average size of Ag nanoparticles in the Ag/chitosan composition?
2. Authors stated that “…, the chitosan film, being hydrophobic, absorbed the solution, causing it to swell and leading to a uniform distribution of the silver precursor within the chitosan film.” Can the hydrophobic chitosan film absorb the solution? Why? It should be a hydrophilic material that absorbs the solution more easily. Water contact angles (WCAs) of the film surfaces should be provided.
3. In the manuscript, How to control the thickness of films and size of Ag nanoparticles?
4. Is the Ag nanparticle size same in different thicknesses films? Please provide evidence.
5. As we know, the different nanparticle size, spacing, shape and quantity all lead to different optical properties. In the manuscript, authors only stated the effect of thin film thickness on optical properties. Do these factors (nanparticle size, spacing, shape and quantity) affect optical performance? The authors should discuss it in detail.
6. In the Introduction, the authors should explain the reason for the introduction of Ag nanoparticles into Chitosan.
7. Do the three-in-one functional properties come from Ag nanoparticles or Chitosan? What role do these two factors play in performance?
Comments on the Quality of English LanguageMinor editing of English language required.
Reviewer 2 Report
Comments and Suggestions for Authors
Some minor typos need correction.
Comments on the Quality of English LanguageA nice paper. Here are some suggestions for improving it:
1) Authors comment on suitability for food packaging and food industry. Materials for this field must comply with rather strict legislation. It is worth adding a sentence on how the new materials comply with such strict rules.
2) In 2.2 I suggest to replace the word "synthesis" with a more adequate word.
3) Authors declare a 1-2 min color change time-scale. I recommend adding a time resolved experiment to show it.
4) In S3 it is clearly demonstrated that the films display a variability in colors. This should be reflected in Fig. 6. of the manuscript by adding error bars.
5) In S4, the right part of the *a axis is in pink. It should be in red...
Reviewer 3 Report
Comments and Suggestions for Authors
In this paper, the authors reported a humidity sensor based on structural color films. The research motivation is interesting, and the results are generally acceptable. But there are some problems in the presentation and discussion of the results. Some modifications are necessary.
1. Introduction: The authors introduced some biocompatible, friendly, and natural materials as well as their multifunctional applications. However, there is a lack of systematic discussion on humidity sensors and humidity sensing materials (traditional resistance, capacitance, voltage (power generation), frequency, etc.), especially humidity induced color changing materials, and may referring to Sens. Actuators, A 2021, 331, 112911; Sens. Actuators, B Chem. 2023, 394, 134445.
2. “In this controlled environment, various RH levels (25, 65, 75, and 90%) were…”. Please provide a schematic diagram of the humidity generation system, as well as corresponding gas flow values and calibration method. Also, why only provide these humidity gradients?
3. Is the discoloration caused by humidity reversible? Can authors provide relevant results?
4. Can the discoloration caused by humidity be directly observed by the human eye? If possible, please provide optical photos, like Sens. Actuators, A 2021, 331, 112911.
5. Is the discoloration caused by humidity affected by temperature?
6. Please provide a humidity sensing mechanism diagram for easy understanding.
Round 2
Reviewer 1 Report
Comments and Suggestions for Authors
In order to exclude the effect of particle size on optical properties, the particle size should be kept constant in the different thickness composites. Thus authors must provide the average size of Ag nanoparticles in the different thickness composites.
Comments on the Quality of English LanguageIn order to exclude the effect of particle size on optical properties, the particle size should be kept constant in the different thickness composites. Thus authors must provide the average size of Ag nanoparticles in the different thickness composites.
Reviewer 3 Report
Comments and Suggestions for Authors
The response and revised manuscript are satisfactory, and it is recommended to accept.
Author Response
We appreciate the reviewer's comprehensive evaluation of our revised manuscript, "Chitosan-Based Structural Color Films for Humidity Sensing with Antiviral Effect." We were pleased to know that the reviewer has found our revised manuscript satisfactorily modified and suitable for publication. We thank the reviewer for taking the time to review our work and provide constructive comments.